# QTL Mapping of Yield, Agronomic, and Nitrogen-Related Traits in Barley (*Hordeum vulgare* L.) under Low Nitrogen and Normal Nitrogen Treatments

**DOI:** 10.3390/plants13152137

**Published:** 2024-08-01

**Authors:** Bingjie Chen, Yao Hou, Yuanfeng Huo, Zhaoyong Zeng, Deyi Hu, Xingwu Mao, Chengyou Zhong, Yinggang Xu, Xiaoyan Tang, Xuesong Gao, Jian Ma, Guangdeng Chen

**Affiliations:** 1College of Resources, Sichuan Agricultural University, Chengdu 611130, China; chenbingjie@stu.sicau.edu.cn (B.C.); houyao6723@163.com (Y.H.); huoyf9918@163.com (Y.H.); zhaoyongzeng0406@163.com (Z.Z.); hudeyi18@126.com (D.H.); maoxingwu@stu.sicau.edu.cn (X.M.); xuyinggang2021@126.com (Y.X.); tangxycau@126.com (X.T.); gxs80@126.com (X.G.); 2College of Economics, Hunan Agricultural University, Changsha 410125, China; youzy13013@163.com; 3Triticeae Research Institute, Sichuan Agricultural University, Chengdu 611130, China; jianma@sicau.edu.cn

**Keywords:** barley, yield, agronomic, N-related traits, QTL mapping

## Abstract

Improving low nitrogen (LN) tolerance in barley (*Hordeum vulgare* L.) increases global barley yield and quality. In this study, a recombinant inbred line (RIL) population crossed between “Baudin × CN4079” was used to conduct field experiments on twenty traits of barley yield, agronomy, and nitrogen(N)-related traits under LN and normal nitrogen (NN) treatments for two years. This study identified seventeen QTL, comprising eight QTL expressed under both LN and NN treatments, eight LN-specific QTL, and one NN-specific QTL. The localized C2 cluster contained QTL controlling yield, agronomic, and N-related traits. Of the four novel QTL, the expression of the N-related QTL *Qstna.sau-5H* and *Qnhi.sau-5H* was unaffected by N treatment. *Qtgw.sau-2H* for thousand-grain weight, *Qph.sau-3H* for plant height, *Qsl.sau-7H* for spike length, and *Qal.sau-7H* for awn length were identified to be the four stable expression QTL. Correlation studies revealed a significant negative correlation between grain N content and harvest index (*p* < 0.01). These results are essential for barley marker-assisted selection (MAS) breeding.

## 1. Introduction

Nitrogen (N) is one of the essential nutrients for plant growth [1]. In agricultural systems, increased application of N fertilizer is the main reason for the rapid increase in crop yields [2]. However, the over-application of N fertilizer increases the cost of agricultural production [3] and results in a series of environmental problems [4,5,6,7]. Relevant studies have revealed significant genotypic differences in crop adaptation to low-nitrogen (LN) stress, with varying N efficiency characteristics [8]. Therefore, there is an urgent need to understand the genetic basis of LN tolerance and breed LN tolerant crop varieties to reduce N fertilizer use, increase crop yields, and promote sustainable agricultural development.

Nutrition-related traits of N metabolism are complex quantitative traits with extensive genetic variation among different genotypes. Quantitative trait locus (QTL) mapping is a very effective method for the genetic dissection of quantitative traits [9]. It has become a routine method for identifying genes or genomic regions that control polygenic traits [10]. Related studies have identified QTL associated with LN tolerance in many plants, including maize [11,12,13,14], rice [15,16,17,18], wheat [19,20,21,22], and Arabidopsis [23]. These studies have identified some yield, agronomic, and N-related traits in response to LN stress [13,16].

Improving nitrogen use efficiency (NUE) is one of the goals of modern agriculture, but it also faces great challenges [24,25]. NUE is a complex trait that depends on two main factors: nitrogen uptake efficiency (NUpE) and nitrogen utilization efficiency (NUtE) [26,27]. Given that complete control over the total amount of nitrogen (N) in field environments is unfeasible, this study concentrates on NUtE. Studies on major grain crops like wheat and rice have demonstrated correlations between N-related traits at crop maturity and agronomic traits (yield) to varying extents [28,29]. Moreover, the correlated QTL for traits also exhibits clustering (tight interlocking) on chromosomes [22,30,31].

Barley (*Hordeum vulgare* L.), the fifth largest cereal crop globally, is used for brewing, foraging, and food. Genotypic differences in barley N-related traits have been demonstrated [32,33]. Previous research has focused on identifying QTL associated with LN tolerance in barley [10,34,35,36,37,38,39,40,41,42]. In addition, QTL linked to barley yield-related traits have been mapped to different chromosomes [40,41,43,44,45]. For example, QTL for harvest index (HI) was localized on 3H and 4H in seven environments using a barley recombinant inbred line (RIL) population [46]. In addition, the QTL for thousand-grain weight (TGW), spike length (SL), and grain number per spike (GN) were localized on 2H, 5H, and 7H under varying environmental conditions [47]. However, these studies lacked N treatments, hindering a comprehensive understanding of the relationship between NUE and yield traits [10,34,35,36,43,44,45]. Although a few studies have localized barley yield traits by different N treatments, no N-related traits have been localized [41]. To further understand the link, it is necessary to localize yield and N-related traits.

Over a century of intensive breeding has led to modern barley cultivars that exhibit limited genetic diversity [48]. However, landraces or wild barley exhibit higher allelic diversity and are better adapted to adverse conditions [49,50]. Our team has investigated related traits in wild barley and identified numerous QTL [41,42]. To maximize the detection of QTL different from wild barley, we selected a landrace barley CN4079 and cultivar Baudin as parents to construct populations for this study. The population was phenotyped, and QTL localization was analyzed for yield, agronomic, and N-related traits at maturity by LN and normal N (NN) treatments. This study aims to identify QTL or QTL clusters that are stably expressed under various N treatments to assess the correlation among yield, agronomic, and N-related traits and to mine some candidate genes. In addition, the study also hopes to further identify alleles conferring tolerance to LN stress and to provide crucial information for marker-assisted selection (MAS) breeding in barley.

## 2. Results

### 2.1. Phenotypic Differences and Analysis

Building on our previous studies on wild barley [41,42], this study investigated twenty traits of three types (Table 1). Among them, yield traits directly correlate with crop yield and its components, agronomic traits are crucial for agricultural productivity, and N-related traits indicate plant NUE.

The parents of the RIL population, Baudin and CN4079, exhibited significant differences (*p* < 0.01) in most of the investigated traits under both LN and NN treatments (Table 2). In both N treatments, Baudin exhibited significantly higher SN and TGW compared to CN4079, while GN was notably lower (*p* < 0.01). Over two years at the LN level, both GDW and ADW of the parental lines and RIL populations realized an increasing trend. Notably, Baudin was consistently higher than CN4079 in GDW and ADW.

The transgressive segregation within the RIL population was significant during the two experimental years, with greater variation in each trait. The frequency distribution showed a continuous normal distribution, indicating that these were typical quantitative traits (Figure 1). In both the parental lines and the RIL population, LN treatment significantly decreased the GDW, StDW, ADW, SN, PH, SL, GP, GNC, StNC, GNA, StNA, and TNA. However, three N-related traits (NHI, NUtE_DM_, and NUtE_GY_) showed significant increases under the same conditions.

### 2.2. Correlation Analysis between Investigated Traits

The results of the correlation analysis between yield-related traits and agronomic traits are shown in Figure 2. The correlation coefficients of the seven yield-related traits with five agronomic traits ranged from −0.88 to 0.95 over two years. PH, AL, LDR, and GP were significantly positively correlated with most of the yield traits within two years. For example, LDR was significantly positively correlated with GN (*p* < 0.05). While SL showed a low correlation with yield-related traits, this correlation was not prone to environmental influences.

Correlation analysis of seven yield-related traits and eight N-related traits indicated that the correlation coefficients of the two main categories of traits ranged from −0.99 to 0.99 (Figure 3). GDW, StDW, ADW, and GNA showed a significant positive correlation in N-related traits (*p* < 0.05). TNA and StDW, TNA and ADW, and NHI were significantly positively correlated with HI (*p* < 0.05). NUtE_DM_ and NUtE_GY_ showed a significant positive correlation with HI at the *p* < 0.01 level. Additionally, significant negative correlations were observed between StNC and HI (*p* < 0.05) as well as between GNC and HI (*p* < 0.01).

The correlation coefficients between five agronomic traits and eight N-related traits were analyzed (Figure 4), ranging from −0.93 to 0.88. Notably, GNC showed a positive correlation with all agronomic traits except AL. Furthermore, StNC, GNA, StNA, and TNA were positively correlated with the examined agronomic traits. However, NHI, NUtE_DM_, and NUtE_GY_ were more affected by the environment and showed varying correlations with the agronomic traits.

### 2.3. QTL Mapping

In this study, seventeen QTL were detected on five barley chromosomes and were associated with twelve investigated traits under different N treatments (Table 3 and Figure 5). Specifically, three, ten, and four QTL were detected to be associated with yield traits (HI, GN, TGW), agronomic traits (PH, SL, AL, LDR, GP), and N-related traits (StNA, NHI, NUtE_DM_, NUtE_GY_), respectively. Eight QTL were exclusively detected in the LN treatment, one QTL was detected only in the NN treatment, and eight QTL were detected at both N levels. Out of seventeen QTL, eight exhibited positive additive effects, with Baudin enhancing these effects, while nine showed negative additive effects, with CN4079 increasing their effects. All QTL contributed between 17.7% and 49.0%, with LOD values ranging from 3.09 to 10.53, indicating their effectiveness.

Four of the eight QTL were detected under both N treatments in both years (Table 3 and Figure 5), including four traits (TGW, PH, SL, AL). *Qtgw.sau-2H* was mapped to the *bpb9984-bpb8143* molecular marker interval on chromosome 2H, which controls TGW. *Qph.sau-3H* was located in the *bpb7245-bpb4616* interval on chromosome 3H, which regulates PH. *Qsl.sau-7H* was located on chromosome 7H within the *bpb5562-bpb7875* interval, which controls SL, while *Qal.sau-7H* controls AL, located on chromosome 7H within the *bpb6214-bpb4441* interval. The additive effects were positive for all except *Qph.sau-3H*, suggesting that the increase in the QTL effect was attributed to Baudin. It is worth noting that the PVE ranges for *Qtgw.sau-2H*, *Qph.sau-3H* and *Qsl.sau-7H* were 41.4–48.7%, 32.8–49.0% and 29.0–45.9%, respectively. Therefore, these three QTL may contain genes with highly significant effects on TGW, PH, and SL.

Except for *Qldr.sau-5H.1*, which was detected only at the NN level, eight QTL were exclusively detected in the LN treatment (Table 3 and Figure 5). Two of these QTL were stably expressed under two years of LN treatment, *Qhi.sau-5H*, which controls HI, and *Qsl.sau-3H*, which controls SL. The remaining six QTL (*Qsl.sau-6H*, *Qal.sau-2H*, *Qldr.sau-3H*, *Qgp.sau-5H.1*, *Qnute_DM_.sau-5H,* and *Qnute_GY_.sau-7H*) were only detected after one-year of LN levels. These LN-specific QTL explanations accounted for 17.7% to 29% of the phenotypic variance and might play a crucial role in LN stress responses.

### 2.4. QTL Cluster

Two QTL clusters, C1 and C2, each comprising three or more QTL, were localized on chromosome 5H (Appendix A), involving seven QTL. The C1 cluster contained three loci, each controlling LDR, GP, and NUtE_DM_, respectively. Among them, *Qldr.sau-5H.1* was detected under E1 NN treatment, with the gain allele originating from the parent CN4079. Meanwhile, *Qgp.sau-5H.1* and *Qnute_DM_.sau-5H* were identified under E1 LN treatment, with the gain allele originating from the parent Baudin. The C2 cluster contained four loci (HI, LDR, StNA, NHI), controlling yield, agronomic, and N-related traits and the gain allele derived from the parent CN4079. Among them, *Qldr.sau-5H.2*, *Qstna.sau-5H*, and *Qnhi.sau-5H* were all detected under both N treatments for one year, whereas *Qhi.sau-5H* demonstrated stable expression under two years of LN treatment.

## 3. Discussion

### 3.1. Novel QTL for Yield, Agronomic, and N-Related Traits

The localization of barley N-related traits has been reported in previous studies [37,39,42,51]. Kindu et al. [37] utilized barley RILs to localize N-related traits across various N levels, including QTL for grain NUE and biomass NUE located on chromosome 5H and QTL for NHI on chromosome 7H [37]. Similarly, Zeng et al. [42] identified QTL for NHI, NUtE_DM_, and NUtE_GY_ traits on chromosomes 2H and 3H using RILs derived from cultivated and wild barley. Han et al. [38] also detected QTL for NUE on 3H. Furthermore, numerous QTL for barley leaf, grain, and stalk N content have been identified by different researchers using various RIL populations and environmental conditions [36,45,52]. In this study, we identified four QTL associated with N-related traits in barley, three of which (*Qstna.sau-5H*, *Qnhi.sau-5H*, *and Qnute_GY_.sau-7H*) were newly discovered and did not overlap with previously reported regions. Furthermore, the expression of *Qstna.sau-5H* and *Qnhi.sau-5H* remained unaffected by N treatments, explaining 25–31.1% and 23.1–26.2% of the phenotypic variation, respectively, indicating the need for further exploration (Table 3 and Figure 5).

Numerous studies have investigated barley yield traits using diverse environmental conditions and RIL populations [43,44,45,47]. Specifically, the localization of thirteen barley traits across six environments revealed that the QTL for TGW was mapped to chromosomes 2H, 4H, 5H, and 7H, while the QTL for HI was mapped to chromosomes 4H, 5H, and 7H [43]. In addition, the QTL for the GN trait was localized on chromosome 2H, utilizing the barley RIL population under different N treatments [40]. This study identified three yield-related QTL (*Qhi.sau-5H*, *Qgn.sau-2H*, and *Qtgw.sau-2H*), all of which overlapped with previously reported regions, validating the accuracy of this study (Table 3 and Figure 5).

Many studies have found that barley agronomic traits QTL have been mapping to different chromosomes [36,37,41,43,53]. For instance, the QTL for SL traits were localized on chromosomes 3H and 6H and the QTL for AL traits were localized on chromosome 7H, utilizing an RIL population constructed from cultivars and wild barley [41]. Furthermore, the QTL for GP and LDR traits were localized on chromosome 5H under different environments [43]. QTL for PH and LDR traits on chromosome 3H were identified, utilizing diverse barley RIL populations and environments [10,54]. In this study, nine of the ten agronomic trait QTL were confirmed to overlap with the regions localized by previous researchers, with the inclusion of one novel QTL (*Qal.sau-2H*) (Table 3 and Figure 5).

### 3.2. NN-Specific and LN-Specific Expression QTL

Many previous studies have shown that research on LN tolerance in crops should be conducted in environments that include LN fertilization rather than only under standard N fertilization treatment for experimental trait phenotypic inference [35,36]. The genetic mechanisms of crops exhibit variability at different N levels. In the previous QTL localization for barley yield, agronomic, and N-related traits at different N levels, loci specifically expressed at the LN level were identified [36,37]. In this study, eight LN-specific QTL and one NN-specific expression QTL (*Qldr.sau-5H.1*) were detected (Table 3 and Figure 5), indicating that the expression of these QTL is susceptible to external N fertilization treatments. The eight LN-specific expression QTL included one yield-related trait locus (*Qhi.sau-5H*), five agronomic traits loci (*Qsl.sau-3H, Qsl.sau-6H, Qal. sau-2H, Qldr.sau-3H, Qgp.sau-5H.1*), and two N-related traits loci (*Qnute_DM_.sau-5H, Qnute_GY_.sau-7H*). Notably, the two QTL within the three traits associated with N efficiency (NHI, NUtE_DM_, and NUtE_GY_) were identified as LN-specific QTL, which might explain the significant increase in phenotypic values under LN input.

The physical intervals of the QTL were targeted using the BLAST function of the BARLEY database to determine the candidate genes (CGs). NAC, BHLH transcription factors, and B-box zinc finger family protein-related CGs were predicted in four of these specific QTL (*Qldr.sau-5H.1, Qldr.sau-3H, Qgp.sau-5H.1, Qnute_GY_.sau-7H*) (Appendix A). NAC transcription factors are associated with nutrient remobilization and increased seed protein content in wheat and barley [55,56], while BHLH transcription factors are engaged in the transport of NH_4_^+^ [57]. Notably, only two CGs were predicted, one BHLH CG and one N high-affinity transporter on *Qnute_GY_.sau-7H*. In addition, CGs related to the B-box zinc finger family protein, which is overexpressed in Arabidopsis and significantly enhances tolerance to abiotic stresses through ABA signaling, were predicted within these four QTL [58].

### 3.3. Stable Expression of QTL at Different N Levels

Different environments or treatments can affect the expression level of genes, ultimately leading to different QTL localization results for the same trait. Many previous studies have indicated that the improvement of LN-tolerant varieties of crops under LN application is worthy of selection rather than inferring phenotypic traits from experiments conducted only under standard N treatment [35,36,59,60,61]. Differences in the genetic mechanisms of barley at different N levels suggested that certain QTL are adapted to specific N levels [35,37]. Therefore, stable QTL and N-specific QTL should be detected at different N levels. Four of the eight QTL under both N treatments were detected in both years (*Qtgw.sau-2H*, *Qph.sau-3H*, *Qsl.sau-7H*, *Qal.sau-7H*), suggesting that they were not (or were not fully) affected by the external N levels (Table 3 and Figure 5). Four stable and consistently expressed QTL identified in this study represent promising candidates for MAS in barley breeding.

The significant effects of different external N levels on barley yield and yield components have been reported [62,63]. Studies in rice and wheat have previously mapped QTL clusters on chromosomes associated with N-related, agronomic, and yield traits [29,64]. In our investigation, we similarly observed clustering, notably the C2 cluster on chromosome 5H containing QTLs controlling HI, LDR, StNA, and NHI, which covered all types of traits (Appendix A). It was demonstrated that the genetic mechanisms among yield, agronomic, and N-related traits were closely related under different N treatments, which was consistent with the results of previous studies [64,65].

We predicted asparagine and glutamate CGs (HORVU.MOREX.r3.2HG0171720.1, HORVU.MOREX.r3.2HG0189840.1, HORVU.MOREX.r3.2HG0202350.1, HORVU.MOREX.r3.3HG0229730.1, HORVU.MOREX.r3.3HG0232400.1, HORVU.MOREX.r3.3HG0287730.1) in two stably expressed QTL (*Qtgw.sau-2H*, *Qph.sau-3H*) (Appendix A). Asparagine and glutamate-related genes were categorized as amino acid biosynthesis classes with the potential to improve plant NUE [38]. For example, asparagine synthase and asparaginase have been reported to affect N utilization and yield in Arabidopsis [66,67]. Glutamate synthase is a key enzyme in ammonium N assimilation [68].

### 3.4. Correlation Analysis of LN Tolerance Traits in Barley

The level of N supply significantly affects agronomic and N-related traits in crops due to the importance of N nutrition for plant growth [34]. For example, N deficiency can reduce pH [10,35,69], shorten GP [11], reduce GY [16], and reduce GNC and StNC [36]. In this study, twenty LN tolerance evaluation indexes of barley were investigated. The results showed that GDW, StDW, ADW, SN, PH, SL, GP, GNC, StNC, GNA, StNA, and TNA traits were significantly reduced by LN treatment. Conversely, three N-related traits (NHI, NUtE_DM_, and NUtE_GY_) showed significant increases (Table 2). These results are consistent with previous studies. Among the three yield factors (SN, GN, and TGW), SN decreased most significantly in LN treatments between the two years, but GN and TGW of the population were less affected by LN stress (Table 2). In addition, other studies have found that TGW in barley is almost unaffected by N fertilization levels [10,34,70], which is also consistent with the results of the present study. Due to the complexity of physiological phenotypic traits, there is no standardized index for evaluating crop tolerance to LN stress. Muurinen et al. [71] showed that NUtE can be assessed by measuring total plant biomass and yield traits. Yield-related traits, biomass, days of tasseling, and plant height are frequently found in QTL localization for LN tolerance in barley [10,34,36,37]. In the present study, positive correlations were observed between the three categories of yield, agronomic, and N-related traits, with the majority reaching significant levels at both N levels (Figure 2, Figure 3 and Figure 4). Among them, there were positive or significant positive correlations between GDW and five N-related traits (GNC, GNA, StNC, StNA, and TNA). This finding is consistent with previous studies [26,52].

## 4. Materials and Methods

### 4.1. Plant Materials

The experimental population consisted of ninety-two RILs from the Baudin × CN4079 crossed over nine generations using the single-grain transmission method. The parent Baudin is a cultivated barley widely grown around the world. The parent CN4079 is a landrace, originating from the Asian region.

### 4.2. Experimental Design

The ninety-two RILs and two parents were planted during 2018–2019 (E1/trial 1) and 2021–2022 (E2/trial 2) at the trial site located in Shifang City, Sichuan Province, China (31°07′ N, 104°24′ E, altitude 542 m), which belongs to the mid-latitude subtropical humid climatic zone, and the cropping system was based on drought and flood crop rotation. The soil type of the experimental field was tidal soil, with an organic matter content of 16.94 g kg^−1^, total nitrogen of 1.25 g kg^−1^, alkaline-dissolved nitrogen of 53.55 mg kg^−1^, quick-acting phosphorus of 13.64 mg kg^−1^, quick-acting potassium of 81.83 mg kg^−1^, and a pH value of 5.52.

The site was categorized into LN level (pure N, 0 kg ha^−1^) and NN level (pure N, 150 kg ha^−1^). In addition, 75 kg ha^−1^ of pure phosphorus and 75 kg ha^−1^ of pure potassium were used for both LN and NN treatments. The three fertilizers were urea, superphosphate, and potassium chloride. The field trials were conducted using a split-plot design, with the primary zone for N levels and the secondary zone for genotypes. The ninety-two RILs and two parents were planted in the same plots with three replications per treatment. Each row was one meter long with a twenty centimeter spacing between rows. Each row had eight seeds and was replicated every three rows, with adjacent materials being spaced one row apart as a protection row. Field management followed local conventional methods.

### 4.3. Trait Measurements and Data Analysis

A total of twenty traits of three types were investigated in this study (Table 1). The PH, SL, AL, LDR, and GP of three plants of each strain were randomly fixed before harvesting at maturity. The LDR was recorded in five levels and studied according to the Agricultural Industry Standard of the People’s Republic of China (NY/T 1301-2007) [72]. Aboveground parts of the three plants in each replicated area were randomly harvested from each strain at maturity, divided into seed and stem components, and dried at 75 °C to a constant weight. Biomass and yield-related traits were also determined. Finally, mixed samples from one replicate were ground and digested using H_2_SO_4_-H_2_O_2_ until the mixture was clarified, and the N content was determined using the Kjeldahl method.

Since the total amount of nutrient elements in the field environment could not be precisely controlled, this study focused only on NUE and N uptake efficiency was not explored. Calculation formula: harvest index = dry weight of seeds/dry weight of aboveground parts [73]; elemental accumulation = elemental content × part biomass [73]; total elemental accumulation = elemental accumulation of seeds + elemental accumulation of stalks [73]; element harvest index = seed grain element accumulation/total element accumulation × 100 [73]; elemental seed production efficiency = seed grain dry weight/total element accumulation [22]; elemental dry matter production efficiency = aboveground dry weight/total elemental accumulation [22].

Analysis of variance (ANOVA), least significant difference (LSD) tests, and correlation analyses (r) between different traits and tests were performed using SPSS version 22.0. Graphics were produced using Origin Pro 8.5 (Origin Lab Corporation, Northampton, MA, USA).

### 4.4. QTL Mapping

The population genetic map for “Baudin × CN4079” RILs was constructed following the methodology outlined in the previous study [74]. DNA isolation was performed using the method described by Chen et al. [74]. DArT genotyping was conducted by Triticarte Pty. Ltd, Canberra, ACT, Australia. (https://www.diversityarrays.com/ (accessed on 10 May 2024)). A genetic map for the population was generated using JoinMap 4.0 software [75]. The total chromosome length of the genetic map was 821 cM, covered by 488 DArT markers, with an average distance of 1.68 cM between markers. The constructed genetic map was compared with existing barley genetic maps in the Grain Genes database (https://wheat.pw.usda.gov/GG3/genome_browser (accessed on 14 July 2024)). There are advantages such as a high number of markers, high density, and small average interval distance, which are advantageous for subsequent QTL mapping.

QTL mapping was performed using the MapQTL6.0 program/MapQTL^®^ 6.0 [76]. The Kruskal–Wallis test was used as a preliminary test for the association between markers. Interval mapping (IM) was then used to identify the major QTL. Automated cofactor selection was used to model multiple QTL (MQM) and significantly associated markers were selected as cofactors/automated cofactor selection was used to identify markers as cofactors. Detection thresholds (*p* < 0.01) for putative QTL were defined by/(based on) 1000 alignments for estimation, and a minimum LOD score of 3.0 was selected. Following the previous method [74], chain mapping was performed using MapChart 2.0 software. The QTL were named according to the recommendations of the International Code of Genetic Nomenclature (http://wheat.pw.usda.gov/ggpages/wgc/98/Intro.htm (accessed on 15 April 2022)). We defined QTL clusters as three or more significant QTL with overlapping confidence intervals [31,77,78,79].

## 5. Conclusions

In this study, we identified eight QTL that showed expression under various nitrogen (N) treatments, with four QTL (TGW, PH, SL, and AL) exhibiting stability across two years. Eight LN-specific QTL and one NN-specific QTL were localized. Notably, two novel QTL related to N efficiency (*Qstna.sau-5H* and *Qnhi.sau-5H*) were discovered under different N treatments. On chromosome 5H, seven QTL aggregated into two distinct clusters. The localized C2 cluster contained QTL controlling yield, agronomic, and N-related traits. In addition, we evaluated the relationships among the yield, agronomic, and N-related traits. Among them, the correlation between yield-related traits and N-related traits was the most significant. It was observed that NUtE_DM_ and NUtE_GY_ exhibited a significant positive correlation with HI (*p* < 0.01), while GNC showed a significant negative correlation with HI (*p* < 0.01). These results provided useful information for exploring LN tolerance and improving N efficiency breeding in barley.

## Figures and Tables

**Figure 1 plants-13-02137-f001:**
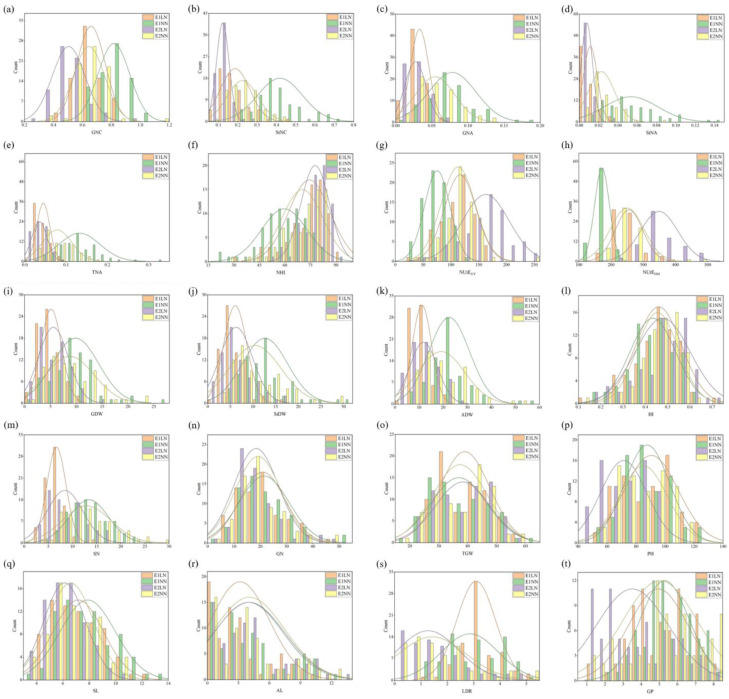
Frequency distribution of twenty traits under different environments and N treatments. GNC, grain nitrogen content (**a**); StNC, stem nitrogen content (**b**); GNA, grain nitrogen accumulation per plant (**c**); StNA, stem nitrogen accumulation per plant (**d**); TNA, total nitrogen accumulation per plant (**e**); NHI, nitrogen harvest index (**f**); NUtE_GY_, N utilization efficiency for grain yield (**g**); NUtE_DM_, N utilization efficiency for aboveground dry matter (**h**); GDW, grain dry weight per plant (**i**); StDW, straw dry weight per plant (**j**); ADW, dry weight for aboveground per plant (**k**); HI, harvest index (**l**); SN, spike number per plant (**m**); GN, grain number per spike (**n**); TGW, thousand-grain weight (**o**); PH, plant height (**p**); SL, spike length (**q**); AL, awn length (**r**); LDR, lodging resistance (**s**); GP, growth period (**t**).

**Figure 2 plants-13-02137-f002:**
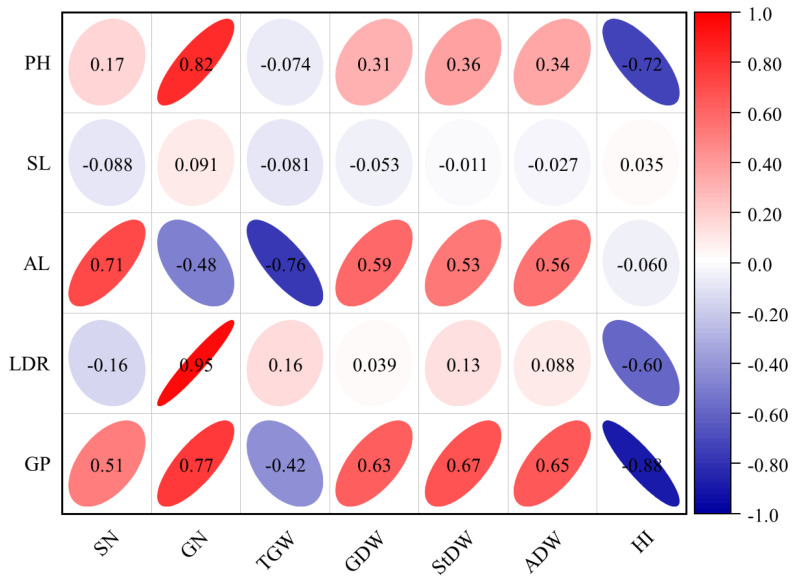
Correlation between agronomic traits and yield-related traits of barley under two nitrogen levels. GDW, grain dry weight per plant; StDW, straw dry weight per plant; ADW, dry weight for aboveground per plant; HI, harvest index; SN, spike number per plant; GN, grain number per spike; TGW, thousand-grain weight; PH, plant height; SL, spike length; AL, awn length; LDR, lodging resistance; GP, growth period. Red represents a positive correlation, blue represents a negative correlation. Shape size denotes the significance of the correlation, with smaller shapes representing stronger correlations.

**Figure 3 plants-13-02137-f003:**
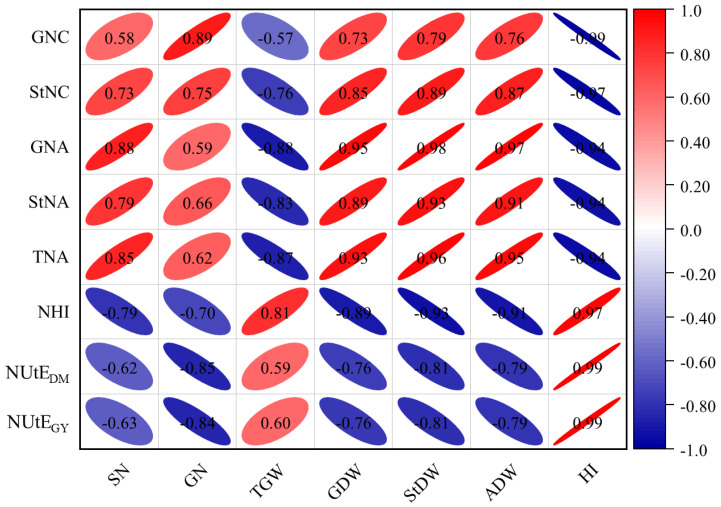
Correlation between N-related traits and yield-related traits of barley under two nitrogen levels. GDW, grain dry weight per plant; StDW, straw dry weight per plant; ADW, dry weight for aboveground per plant; HI, harvest index; SN, spike number per plant; GN, grain number per spike; TGW, thousand-grain weight; GNC, grain nitrogen content; StNC, stem nitrogen content; GNA, grain nitrogen accumulation per plant; StNA, stem nitrogen accumulation per plant; TNA, total nitrogen accumulation per plant; NHI, nitrogen harvest index; NUtE_DM_, N utilization efficiency for aboveground dry matter; NUtE_GY_, N utilization efficiency for grain yield. Red represents a positive correlation, blue represents a negative correlation. Shape size denotes the significance of the correlation, with smaller shapes representing stronger correlations.

**Figure 4 plants-13-02137-f004:**
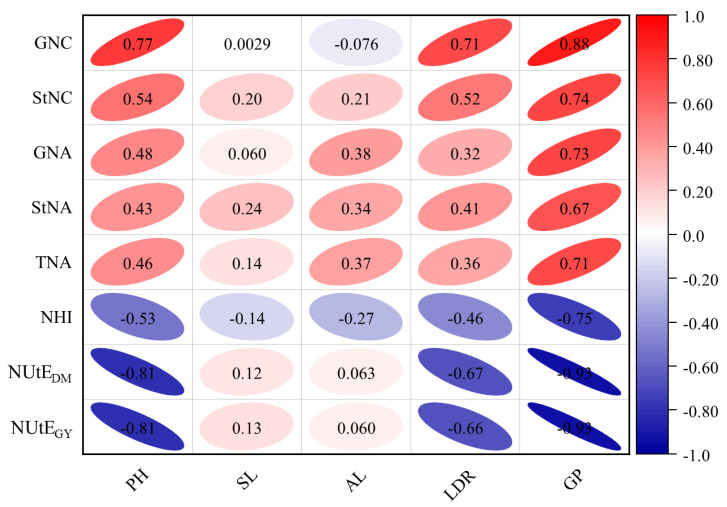
Correlation between N-related traits and agronomic traits of barley under two nitrogen levels. PH, plant height; SL, spike length; AL, awn length; LDR, lodging resistance; GP, growth period; GNC, grain nitrogen content; StNC, stem nitrogen content; GNA, grain nitrogen accumulation per plant; StNA, stem nitrogen accumulation per plant; TNA, total nitrogen accumulation per plant; NHI, nitrogen harvest index; NUtE_DM_, N utilization efficiency for aboveground dry matter; NUtE_GY_, N utilization efficiency for grain yield. Red represents a positive correlation, blue represents a negative correlation. Shape size denotes the significance of the correlation, with smaller shapes representing stronger correlations.

**Figure 5 plants-13-02137-f005:**
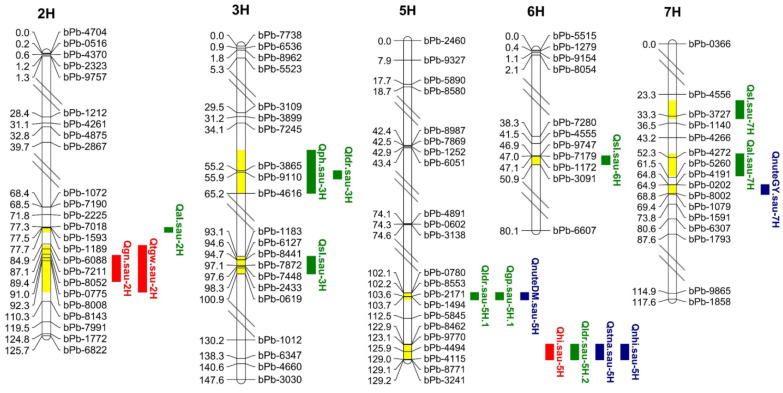
Quantitative trait loci (QTL) for Yield, agronomic traits, and N efficiency at maturity in RIL populations. Yellow represents the relative position of the QTL on the chromosome. Red QTL represents yield-related traits, green QTL represents agronomic traits, and blue QTL represents N utilization traits.

**Table 1 plants-13-02137-t001:** Summary of investigated traits and their measurement methods in this study.

Types	Traits	Abbreviations	Units	Methods of Trait Measurement
Yield traits	Grain dry weight per plant	GDW	g·plant^−1^	
	Straw dry weight per plant	StDW	g·plant^−1^	Dried and weighted using 1/100 balances
	Dry weight for aboveground per plant	ADW	g·plant^−1^	
	Harvest index	HI	-	GDW/ADW
	Spike number per plant	SN	-	
	Grain number per spike	GN	-	
	Thousand-grain weight	TGW	g	Weighted three times of 200 grains for each line in each replication after harvested using 1/1000 balances
Agronomic traits	Plant height	PH	cm	The average value of 10 random individual plants of each line in each replication
	Spike length	SL	cm	
	Awn length	AL	cm	
	Lodging resistance	LDR	-	Visual rating(1–5) of the severity of lodging at maturity, where 1 represents no lodging and 5 represents total lodging
	Growth period	GP	-	
N-related traits	Grain nitrogen content	GNC	%	Automatic Kjeldahl (SKD-100, Peiou Ltd., Shanghai 201108, China)
	Stem nitrogen content	StNC	%	
	Grain nitrogen accumulation per plant	GNA	mg·plant^−1^	GNC × GDW
	Stem nitrogen accumulation per plant	StNA	mg·plant^−1^	StNC × StDW
	Total nitrogen accumulation per plant	TNA	mg·plant^−1^	GNC ± StNC
	Nitrogen harvest index	NHI	%	GNA/TNA × 100
	N utilization efficiency for aboveground dry matter	NUtE_DM_	g·g^−1^	ADW/TNA
	N utilization efficiency for grain yield	NUtE_GY_	g·g^−1^	GDW/TNA

**Table 2 plants-13-02137-t002:** Phenotypic variation in barley parents and RIL population under two nitrogen levels.

Types	Trait	Treatment	E1					E2				
			Parents		RIL Population			Parents		RIL Population		
			Baudin	CN4079	Mean ± SD	Range	CV (%)	Baudin	CN4079	Mean ± SD	Range	CV (%)
Yield traits	GDW	LN	8.41 **	4.88 **	5.10 ** ± 2.00	0.98–10.48	39.35	7.39 **	6.26 **	5.53 ** ± 2.92	1.23–14.26	52.72
		NN	20.19	11.32	9.71 ± 4.38	1.96–27.19	45.08	11.89	15.76	8.58 ± 4.61	1.34–22.97	53.78
	StDW	LN	6.56 **	4.25 **	6.09 ** ± 2.57	1.81–16.48	42.13	5.65 **	5.73 **	6.30 ** ± 3.66	1.35–16.72	58.14
		NN	13.35	9.65	12.68 ± 5.10	4.55–30.04	40.19	10.96	14.91	10.34 ± 5.69	2.96–28.69	54.98
	ADW	LN	14.97 **	9.13 **	11.19 ** ± 3.99	3.34–24.93	35.63	13.04 **	11.98 **	11.84 ** ± 5.77	2.71–26.41	48.75
		NN	33.54	20.97	22.39 ± 8.51	7.31–57.23	37.99	22.86	30.67	18.93 ± 9.69	5.12–51.16	51.17
	HI	LN	0.56 **	0.54	0.46 ± 0.10	0.14–0.70	22.51	0.57 *	0.52	0.48 ± 0.13	0.16–0.74	26.23
		NN	0.6	0.54	0.43 ± 0.10	0.15–0.62	23.62	0.52	0.51	0.45 ± 0.09	0.14–0.61	20.61
	SN	LN	8.0 **	4.8 **	6.7 ** ± 1.97	2.3–14.2	29.48	7.5 **	5.5 **	8.5 ** ± 3.75	2.0–17.0	44.2
		NN	19.8	11.5	13.4 ± 4.03	5.3–23.8	30.25	20	11.5	12.8 ± 5.01	3.0–28.5	39
	GN	LN	22.4 **	34.4 **	20.7 ± 8.49	7.1–47.1	41.1	21.9 **	44.3 **	18 ± 7.91	3.7–45.3	44.01
		NN	21.8	50	21.1 ± 8.98	5–52.8	42.66	18	46	18.7 ± 8.11	3.8–48.3	43.29
	TGW	LN	46.93	29.37 **	39.1 ± 8.53	25.1–58.9	21.83	45.0 **	25.7 **	38.4 ± 9.28	21.3–55.2	24.15
		NN	46.99	26.53	37 ± 9.65	19.2–61.3	26.12	33.2	29.7	37.6 ± 9.46	18.4–56.5	25.13
Agronomic traits	PH	LN	61.46 **	83.23 *	70.47 ** ± 18.58	39.37–109.56	26.37	56.04 *	84.45 *	73.88 ** ± 16.36	44.74–106.02	22.14
		NN	73.05	95.47	87.73 ± 15.76	56.52–127.34	17.96	69.97	97.57	85.04 ± 16.52	53.96–114.72	19.43
	SL	LN	9.73	4.26	7.16 * ± 1.87	3.96–12.44	26.12	6.51	3.54	6.07 ± 1.62	3.60–9.57	26.69
		NN	10.68	4.69	8.17 ± 2.07	3.77–13.18	25.34	7.59	3.6	6.64 ± 1.73	3.43–10.09	26.05
	AL	LN	8.01 *	0	3.13 ± 1.05	0.00–10.72	33.55	10.33	0	3.95 ± 1.14	0.00–12.04	28.86
		NN	9.32	0	3.97 ± 1.21	0.00–12.49	30.48	9.04	0	4.03 ± 1.26	0.00–12.45	31.27
	LDR	LN	2.33	3.67	3.09 ± 0.74	1.00–5.00	23.95	1.67	3.67	3.31 ± 0.88	1.00–5.00	26.59
		NN	1.67	4	2.82 ± 0.86	1.00–5.00	30.5	2	3.67	2.40 ± 0.64	1.00–5.00	26.67
	GP	LN	175.00 **	156.33 **	165.90 ** ± 4.57	156.00–177.33	2.75	169.33 **	157.67 **	163.62 ** ± 5.87	155.33–179.00	3.59
		NN	177.67	158.67	168.94 ± 5.28	157.33–179.67	3.13	173	161.33	167.43 ± 6.69	156.67–180.00	4
N-related traits	GNC	LN	0.52 **	0.64 **	0.66 ** ± 0.10	0.47–0.86	14.85	0.39 **	0.37 **	0.51 ** ± 0.11	0.29–0.94	22.12
		NN	0.65	0.85	0.82 ± 0.11	0.60–1.06	13.42	0.59	0.6	0.65 ± 0.11	0.33–1.15	16.87
	StNC	LN	0.19 **	0.18 **	0.19 ** ± 0.08	0.06–0.50	41.69	0.07 **	0.11 **	0.12 ** ± 0.03	0.07–0.25	26.4
		NN	0.32	0.27	0.42 ± 0.12	0.21–0.74	27.59	0.29	0.21	0.22 ± 0.08	0.08–0.44	35.62
	GNA	LN	43.73 **	31.26 **	33.01 ** ± 12.67	8.15–68.06	38.37	28.89 **	23.13 **	27.37 ** ± 14.83	6.47–74.21	54.17
		NN	130.58	96.63	78.24 ± 36.34	19.43–250.91	46.45	70.45	94.73	53.68 ± 27.10	14.49–137.56	50.48
	StNA	LN	12.29 **	7.47 **	11.51 ** ± 7.28	2.66–35.57	63.27	4.11 **	6.35 **	7.91 ** ± 5.84	1.22–33.00	73.86
		NN	21.48	25.89	54.33 ± 27.62	10.99–147.45	50.83	31.51	31.64	22.62 ± 13.62	6.12–84.48	60.2
	TNA	LN	56.01 **	38.73 **	44.52 ** ± 15.98	16.94–90.12	35.9	33.00 **	29.49 **	35.31 ** ± 18.61	7.74–87.97	52.7
		NN	173.55	122.52	132.57 ± 53.96	39.94–398.26	40.7	101.96	126.37	77.17 ± 34.54	28.34–179.33	44.76
	NHI	LN	78.06 *	80.72 *	74.26 ** ± 12.36	32.52–91.88	16.63	87.54 **	78.46	77.64 ** ± 8.37	53.19–90.24	10.78
		NN	75.24	78.87	59.12 ± 13.24	20.61–82.31	22.4	69.1	74.97	70.41 ± 11.96	28.24–84.44	16.98
	NUtE_DM_	LN	267.17 **	235.75 **	255.68 ** ± 43.07	166.06–363.39	16.85	395.04 **	406.31 **	347.63 ** ± 58.03	233.00–499.27	16.69
		NN	193.27	171.16	171.45 ± 23.33	130.48–248.45	13.6	224.16	242.67	249.77 ± 51.94	173.13–477.71	20.79
	NUtE_GY_	LN	150.12 **	126.01 **	116.08 ** ± 28.66	39.20–167.48	24.69	223.97 **	212.15 **	161.26 ** ± 41.15	56.65–249.89	25.52
		NN	116.34	92.42	74.58 ± 22.76	20.79–128.66	30.52	116.64	124.72	112.61 ± 30.47	31.23–244.24	27.06

LN: low-nitrogen stress; NN: normal-nitrogen treatment; E1 and E2 specifically refer to the field trial of barley mature years. GDW, Grain dry weight per plant; StDW, Straw dry weight per plant; ADW, Dry weight for aboveground per plant; HI, Harvest index; SN, Spike number per plant; GN, Grain number per spike; TGW, Thousand-grain weight; PH, Plant height; SL, Spike length; AL, Awn length; LDR, Lodging resistance; GP, Growth period; GNC, Grain nitrogen content; StNC, Stem nitrogen content; GNA, Grain nitrogen accumulation per plant; StNA, Stem nitrogen accumulation per plant; TNA, Total nitrogen accumulation per plant; NHI, Nitrogen harvest index; NUtE_DM_, N utilization efficiency for aboveground dry matter; NUtE_GY_, N utilization efficiency for grain yield. Values followed by different letters are significantly different at *p* < 0.01 between two parents in the same nitrogen level. * indicated a significant difference at *p* < 0.05 for the parents and RIL between the two nitrogen treatments, ** mean significant difference at *p* < 0.01 for the parents and RIL between the two nitrogen treatments. CV: coefficient of variation.

**Table 3 plants-13-02137-t003:** Summary of QTL detected under two nitrogen levels.

Types	Trait	QTL	Chr	Marker Interval	LOD	PVE (%)	Origin	Treatment
Yield traits	HI	*Qhi.sau-5H*	5H	*bpb8462-bpb3241*	5.42	29	CN4079	E1LN
					3.88	22.3	CN4079	E2LN
	GN	*Qgn.sau-2H*	2H	*bpb6088-bpb0858*	4.3	24.4	CN4079	E1LN
					6.19	32.7	CN4079	E1NN
	TGW	*Qtgw.sau-2H*	2H	*bpb9984-bpb8143*	10.4	48.7	Baudin	E1LN
					9	43.4	Baudin	E1NN
					7.93	41.4	Baudin	E2LN
					9.11	45.1	Baudin	E2NN
Agronomic traits	PH	*Qph.sau-3H*	3H	*bpb7245-bpb4616*	9.06	44	CN4079	E1LN
					6.31	32.8	CN4079	E1NN
					10.53	49	CN4079	E2LN
					8.45	42.6	CN4079	E2NN
	SL	*Qsl.sau-3H*	3H	*bpb1183-bpb0619*	3.46	19.8	Baudin	E1LN
					4.48	24.9	Baudin	E2LN
		*Qsl.sau-6H*	6H	*Bpb9839-bpb3091*	3.66	20.9	CN4079	E2LN
		*Qsl.sau-7H*	7H	*bpb5562-bpb7875*	9.29	44.8	Baudin	E1LN
					5.44	29	Baudin	E1NN
					6.7	34.9	Baudin	E2LN
					9.32	45.9	Baudin	E2NN
	AL	*Qal.sau-2H*	2H	*bpb9984-bpb1189*	3.48	20	Baudin	E1LN
		*Qal.sau-7H*	7H	*bpb6214-bpb4441*	4.64	25.7	Baudin	E1LN
					6.58	34	Baudin	E1NN
					6.37	33.4	Baudin	E2LN
					6.9	36.5	Baudin	E2NN
	LDR	*Qldr.sau-3H*	3H	*bpb3865-bpb9110*	3.31	18.9	CN4079	E1LN
		*Qldr.sau-5H.1*	5H	*bpb5179-bpb1494*	3.28	18.5	CN4079	E1NN
		*Qldr.sau-5H.2*	5H	*bpb8462-bpb3241*	5.72	30.6	CN4079	E2LN
					3.92	23	CN4079	E2NN
	GP	*Qgp.sau-5H.1*	5H	*bpb5179-bpb1494*	3.1	17.8	Baudin	E1LN
N-related traits	StNA	*Qs_t_na.sau-5H*	5H	*bpb8462-bpb3241*	5.9	31.1	CN4079	E1LN
					4.62	25	CN4079	E1NN
	NHI	*Qnhi.sau-5H*	5H	*bpb8462-bpb3241*	4.83	26.2	CN4079	E1LN
					4.21	23.1	CN4079	E1NN
	NUtE_DM_	*Qnu_t_e_DM_.sau-5H*	5H	*bpb5179-bpb1494*	3.09	17.7	Baudin	E1LN
	NUtE_GY_	*Qnu_t_e_GY_.sau-7H*	7H	*bpb8002-bpb1079*	3.15	18.7	Baudin	E2LN

LN: low-nitrogen stress; NN: normal-nitrogen treatment; HI, harvest index; SN, spike number per plant; GN, grain number per spike; TGW, thousand-grain weight; PH, plant height; SL, spike length; AL, awn length; LDR, lodging resistance; GP, growth period; StNA, stem nitrogen accumulation per plant; NHI, nitrogen harvest index; NUtE_DM_, N utilization efficiency for aboveground dry matter; NUtE_GY_, N utilization efficiency for grain yield; LOD, the logarithm of odds; PVE, phenotype variance explained.

## Data Availability

All data generated or analyzed during this study are included in this published article and its Appendix A.

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
