# Peer review of "QTL Mapping of Yield, Agronomic, and Nitrogen-Related Traits in Barley (Hordeum vulgare L.) under Low Nitrogen and Normal Nitrogen Treatments"

_plants, 2024, doi:10.3390/plants13152137_

Round 1

Reviewer 1 Report

Comments and Suggestions for Authors

Review on QTL Mapping of yield, agronomic, and nitrogen-related traits in barley (Hordeum vulgare L.) under low nitrogen and normal nitrogen treatments

This manuscript described the QTL analysis of a barley population based on a two-year study at two levels of nitrogen treatment.  This paper is extremely difficult to read due to the excessive use of short forms, lack of proper description experimental methods, very poor data presentation and grammatic errors.   It cannot be published without major revision. 

For the paper to be readable, I suggest to only use shortforms in tables and figures, but full form of traits should be used in the body of text consistently.

Here are points for major revision.

1.     Genetic map construction was briefly described in line 346-347.  Detailed description of map construction is missing and should be added.  In addition, comparison of this map with existing barley genomes and genetic maps (such as https://wheat.pw.usda.gov/GG3/genome_browser) is necessary.  Chromosomal number and marker positions should be blasted against the more widely used barley reference genome for cross-referencing within the barley research community. 

2.     Need to add method of putative candidate gene discovery.

3.     Please add normality and distribution tests of the traits for the population. 

4.     Table 3 needs to move to the beginning of results and provide information of why these authors decided to collect these traits and how the traits are relevant to yield, agronomy and soil nitrogen content.

5.     Table 1 is not readable.  It needs to be reformatted.

6.     The authors need to check for grammatic errors carefully throughout the paper. 

7.     Line 76, “Baudin was better than CN4079 in SN and TGW, whereas CN4079 was better than Baudin in GN and was not affected by N levels.” What does “better” mean.

8.     Line 69 to 70 are fragmented, please revise.

9.      Line 267 “asparagine and glutamate CGs “ What are these genes?  Please use the proper name of the genes.

10.  Line 277 to 280, grammatic error.  Please revise.

Comments on the Quality of English Language

No comment

Reviewer 2 Report

Comments and Suggestions for Authors

The manuscript plants-3065644 entitled "QTL Mapping of yield, agronomic, and nitrogen-related traits in barley (Hordeum vulgare L.) under low nitrogen and normal nitrogen treatments" submitted by Chen et al. report an intersting genetic study related to the QTL mapping of N efficiency related genes.

Considering the importance of this kind of study to improve knowledge regarding the genes involved in the plant N metabolism in order to increase plant NUE and, in particular, about barley that is an important food/fodder product , I believe that the manuscript is of potential interest to readers of “Plants” and falls within its scope.

The experimental activity was carried out following appropriate methods. The experiment was carried out in field and, most important, during two cropping cycle. That allow to obtain high reliable experimental data.

The authors should motivate in details why did they choose to made the RIL from these two cultivars.

The manuscript is well written, easy to read and follow. Only some change of style must to be done. In particular, a lot of acronyms have been used and they should be made explicit before being stated in the text.

My specific comments are detaled in the attached file.

Round 2

Reviewer 1 Report

Comments and Suggestions for Authors

The authors made substantial improvement in the revision.  I recommend consideration of publication.  However, there are still grammatic errors that the authors are responsible to make corrections.

Comments on the Quality of English Language

There are still grammatic errors that the authors are responsible to make corrections.
